# THE GEOMETRY OF GROKKING:
# NORM MINIMIZATION ON THE ZERO-LOSS MANIFOLD

## ABSTRACT

Grokking is a puzzling phenomenon in neural networks where full generalization occurs only after a substantial delay following the complete memorization of the training data. Previous research has linked this delayed generalization to representation learning driven by weight decay, but the precise underlying dynamics remain elusive. In this paper, we argue that post-memorization learning can be understood through the lens of constrained optimization: gradient descent effectively minimizes the weight norm on the zero-loss manifold. We formally prove this in the limit of infinitesimally small learning rates and weight decay coefficients. To further dissect this regime, we introduce an approximation that decouples the learning dynamics of a subset of parameters from the rest of the network. Applying this framework, we derive a closed-form expression for the post-memorization dynamics of the first layer in a two-layer network. Experiments confirm that simulating the training process using our predicted gradients reproduces both the delayed generalization and representation learning characteristic of grokking.

## 1 INTRODUCTION

Neural networks have achieved great success, but their mechanisms remain far from being fully understood. Doshi-Velez & Kim (2017) argue that understanding the inner workings of neural networks is crucial for the development of AI systems with increased safety and reliability. Moreover, understanding the learning dynamics of neural networks could also help us improve their performance and efficiency: it is easier to design better learning algorithms when we understand the limitations of the existing ones. Furthermore, insights into artificial neural networks may also enhance our understanding of biological neural networks due to their fundamental similarities (Sucholutsky et al., 2023; Kohoutová et al., 2020).

This work aims to clarify the learning dynamics underlying a particularly puzzling phenomenon termed *grokking*. Under specific training conditions, neural networks achieve generalization on the test data only after an extended period following the complete memorization of the training data. This behavior was first observed in synthetic problems such as modular addition (Power et al., 2022), but was later shown to also happen in real-world datasets (Liu et al., 2022b; Humayun et al., 2024).

In the specific problem of modular addition, interpretability research has revealed that neural networks achieve generalization by placing the embedding vectors on a circle (Gromov, 2023; Zhong et al., 2024). The circular structure of the embedding layer enables the network to perform a symmetric algorithm that generalizes perfectly to unseen data. It is currently known that circular representations emerge gradually during the post-memorization phase (Nanda et al., 2023) and that weight decay is mainly responsible for driving the delayed generalization (Liu et al., 2022b), but the precise dynamics remain unclear.

Moreover, the role of the embedding layer in the modular addition task is a striking example that generalization in neural networks often hinges on representation learning within specific network components. In such cases, it can be highly beneficial to simplify the complex learning dynamics of a deep network by isolating and analyzing only the component of interest. Recent work has already begun to explore such approximations, also referred to as *effective theories* (van Rossem & Saxe, 2024; Liu et al., 2022a; Musat, 2024; Mehta et al.).

## 2 OUR CONTRIBUTIONS

In this work, we aim to answer the following questions:

**Q1.** What is the exact role of weight decay in the post-memorization learning dynamics?
**Q2.** Can we isolate the dynamics of the embedding layer from the rest of the network?

We answer **Q1** in Section 4 by proving that, after memorization is achieved, the learning dynamics approximately follow the minimization of the weight norm, constrained to the zero-loss level set.

We answer **Q2** in Section 5 by proposing an approximation for the isolated learning dynamics of any parameter subset as the minimization of a specific cost function.

We then combine these insights in Section 6 to study the post-memorization learning dynamics of a two-layer network, deriving a closed-form expression for the cost function of the first layer (the embedding layer).

Finally, in Section 7, we validate our theoretical insights on a modular addition task, showing that our approximations reproduce the delayed generalization and circular representations characteristic of grokking.

## 3 INTUITIONS FROM TOY MODELS

Before presenting our theoretical findings, we give an intuition for our results by discussing and visualizing how they relate to a few highly simplified models.

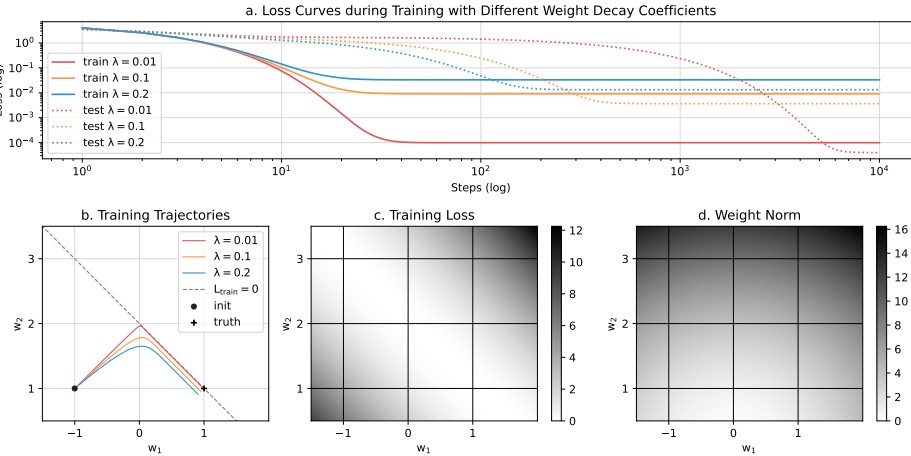

Figure 1: A two-parameter linear model $\hat{y} = w_1 x_1 + w_2 x_2$ groks simple addition when trained with just one sample: $x_1 = x_2 = 1$, $y = 2$ (corresponding to $1 + 1 = 2$). We plot three training runs with different weight decay coefficients $\lambda$. After quickly achieving (almost) zero loss, learning is entirely driven by the minimization of the weight norm.

### 3.1 GROKKING ADDITION

We begin by discussing how a linear model can grok addition from just $1 + 1 = 2$. We use a single-layer linear model with two inputs and two weights: $\hat{y} = w_1 x_1 + w_2 x_2$. We train this model with mean-squared error loss using just one sample: $x_1 = x_2 = 1$, $y = 2$. We use three different values of weight decay, $\lambda \in \{0.01, 0.1, 0.2\}$. We initialize our model with $w_1 = -1$ and $w_2 = 1$.

We aim to show that our model can learn to perform standard addition, despite being trained with a single sample. Test accuracy is measured on a set of 100 randomly generated samples, where $x_1$ and $x_2$ are sampled from a normal distribution, and $y = x_1 + x_2$.

We show our results in Figure 1. We can see that our model reproduces grokking: training loss becomes very low after just a 10 steps, while test loss takes a few hundred steps. Additionally, the model achieves lower loss with smaller $\lambda$, but takes longer to generalize.

**Interpretation.** From Figure 1 (a, b), we can see that learning follows two phases. In the first phase, driven by loss minimization, the model achieves a low loss by learning $(w_1, w_2) \approx (0, 2)$. In the second phase, learning is entirely driven by weight decay. The model follows norm minimization, while maintaining (almost) zero loss, eventually reaching $(w_1, w_2) \approx (1, 1)$. Note that, for smaller $\lambda$, the model remains closer to the zero-loss line, but generalization also takes longer.

### 3.2 NORM MINIMIZATION

While the previous example illustrates our theoretical framework, it does not capture its full generality. It is unsurprising that applying weight decay encourages a reduction in norm. However, the central claim of this paper is significantly stronger: we argue that the learning dynamics under weight decay do not merely follow *some* norm-decreasing direction, but rather evolve along the direction *that maximally decreases the norm, subject to remaining on the zero-loss manifold*. Another way to view this is the following: once the model achieves perfect memorization, learning effectively follows gradient descent on the weight norm, constrained to the zero-loss manifold.

To offer a better intuition, we show how a linear model can grok three-number addition. We train a three-parameter linear model $\hat{y} = w_1 x_1 + w_2 x_2 + w_3 x_3$ with just one sample: $x_1 = x_2 = x_3 = 1$, $y = 3$. As in the previous section, this model exhibits grokking: after quickly achieving zero training loss, the model slowly reaches the generalizing solution $(w_1, w_2, w_3) \approx (1, 1, 1)$. We perform four training runs with different initializations and visualize the resulting trajectories in Figure 2. We observe that, for all initializations, the model first converges to the zero-loss plane, then moves directly towards the solution of minimum norm.

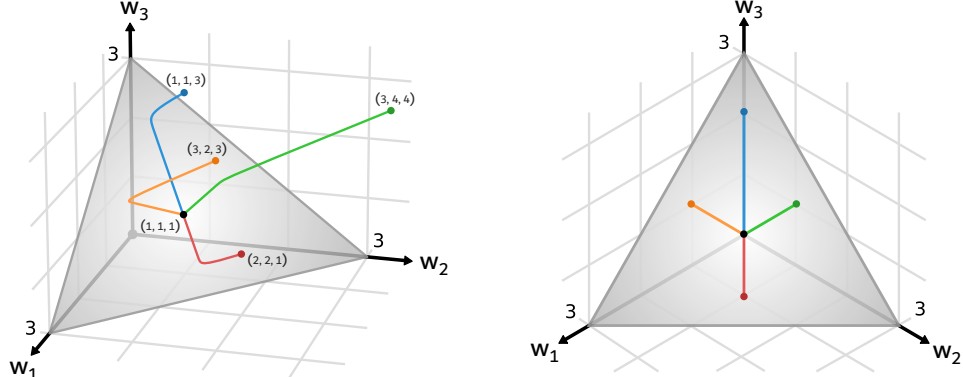

Figure 2: A three-parameter linear model $\hat{y} = w_1 x_1 + w_2 x_2 + w_3 x_3$ groks three-number addition when trained with just one sample: $x_1 = x_2 = x_3 = 1$, $y = 3$ (corresponding to $1 + 1 + 1 = 3$). The gray area shows the zero-loss plane, shaded according to the weight norm, where a lighter shade denotes a lower norm.

### 3.3 A FEW MATHEMATICAL NUANCES

So far, our examples have shown only flat zero-loss subspaces, but this is not necessarily the case. The zero-loss subspace can more generally be thought of as a manifold: a subspace that locally resembles Euclidean space near each point. For example, we show a curved zero-loss set in Figure 3 (left), along with a few training trajectories.

An important caveat is that the zero-loss set is not necessarily a manifold everywhere: it might contain singular points. Such a singular point is demonstrated in Figure 3 (center). However, such singularities should not worry us too much. As we prove in Theorem 4.10, if the network is realized by a smooth function, we will *almost* never encounter a singularity during standard training.

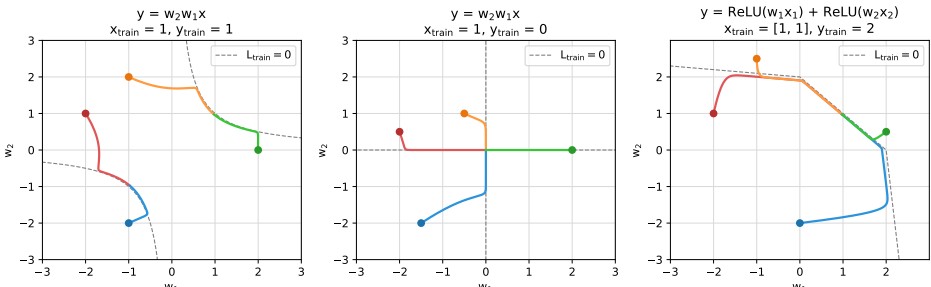

Figure 3: Training trajectories with different data, architectures and initializations. *Left:* a two-layer linear network where the zero-loss set is curved. *Center:* a two-layer linear network where the zero-loss set has a singularity at $(w_1, w_2) = (0, 0)$. *Right:* a single-layer network with leaky ReLU activation groks simple addition.

Another nuance is that, in practice, neural networks are trained using the ReLU activation function, which is not smooth. This will partition the loss into a finite set of smooth regions with nonsmooth boundaries. The nonsmooth points will also form a null set. We visualize a scenario of this type in Figure 3 (right) with leaky ReLU activation: $\text{ReLU}(x) = x$ if $x > 0$ else $x/10$.

### 3.4 Gradient Orthogonality

We further illustrate our key theoretical result using a toy loss landscape. Consider a model with only two parameters $x, y \in \mathbb{R}$ and a loss function $\mathcal{L}(x, y) = (y - x^2)^2$. We visualize this loss landscape in Figure 4. We plot three points $A, B, C \in \mathbb{R}^2$ that have the same projection on the zero-loss manifold, but are getting progressively closer. As we prove in Theorem 4.14, the loss gradients become perfectly orthogonal to the zero-loss set as we approach it.

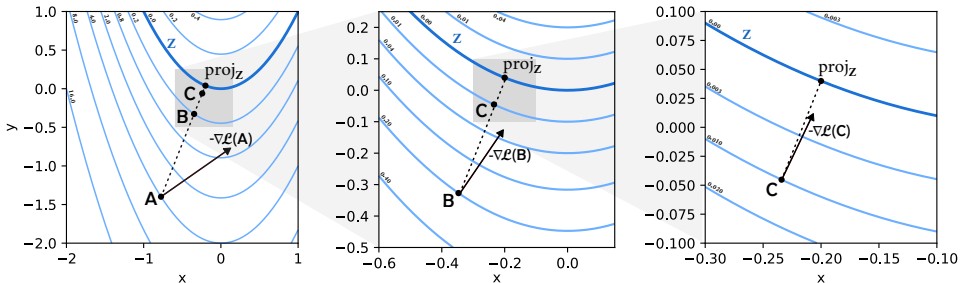

Figure 4: We illustrate Theorem 4.14 using a toy loss landscape $\mathcal{L}(x, y) = (y - x^2)^2$. We plot the level sets of $\mathcal{L}$ from three different views. Each view shows a magnified region of the previous view, zooming into the zero-loss manifold. We display the gradient angle at three different points, pointed in the negative direction. The gradient norm is not shown to scale. Gradients become increasingly orthogonal to the zero-loss manifold as we get closer.

## 4 Post-Memorization Dynamics

### 4.1 Architecture

We consider a neural network trained with mean-squared error loss on $k$ samples and weight decay:

$$\mathcal{L}(\theta) = \sum_{i=1}^{k} \|f(\theta, x_i) - y_i\|^2 \qquad\qquad \mathcal{L}_\lambda(\theta) = \mathcal{L}(\theta) + \lambda\|\theta\|^2 \qquad (1)$$

where $x_i \in \mathbb{R}^n$, $y_i \in \mathbb{R}^m$, and $\theta \in \mathbb{R}^d$ is the parameter vector. The network has $d$ parameters, $n$ inputs, and $m$ outputs. We use $f : \mathbb{R}^{d \times n} \to \mathbb{R}^m$ to denote the network realization function. We apply a weight decay term (Krogh & Hertz, 1991) with a coefficient $\lambda > 0$.

We also denote the concatenated outputs for all training samples using $\mathcal{F} : \mathbb{R}^d \to \mathbb{R}^{km}$, where

$$\mathcal{F}(\theta) = \left[ \; f(\theta, x_1)^\top, \; f(\theta, x_2)^\top, \; \ldots, f(\theta, x_n)^\top \; \right]^\top. \tag{2}$$

### 4.2 THEORETICAL SETUP

We theoretically study the training dynamics under the following assumptions:

**Assumption 4.1** (Over-Parametrization). *We assume that $d \geq km$ in order for the model to be able to memorize the entire dataset without learning any representations.*

**Assumption 4.2** (Smooth Network). *We assume that the network realization function $f$ is continuously differentiable $d - km + 1$ times.*

**Assumption 4.3** (Gradient Flow). *We model the gradient descent trajectory as a gradient flow. We consider the parameter vector as a continuous function of time $\theta : \mathbb{R}_{\geq 0} \to \mathbb{R}^d$ with dynamics:*

$$\frac{\partial \theta(t)}{\partial t} = -\nabla \mathcal{L}_\lambda(\theta(t)). \tag{3}$$

**Assumption 4.4** (Perfect Memorization). *We study the training dynamics after perfect memorization is achieved. This requires that the zero-loss set is not empty, i.e. $\mathcal{Z} \neq \varnothing$.*

**Assumption 4.5** (Vanishing Weight Decay). *We study the learning dynamics in the approximation of a very small weight decay coefficient, i.e. $\lambda \to 0$, motivated by the small values of $\lambda$ typically used by practitioners (Smith, 2018), as well as by previous empirical works on grokking (Liu et al., 2022b) suggesting that a small $\lambda$ is essential for the delayed-generalization phenomona.*

### 4.3 CONSTRAINED TO THE ZERO-LOSS SET

We begin by establishing that the model remains constrained arbitrarily close to the zero-loss set after reaching a memorizing solution.

**Definition 4.6** (Zero-Loss Set). *Let $\mathcal{Z} = \left\{ \theta \in \mathbb{R}^d \,\middle|\, \mathcal{L}(\theta) = 0 \right\}$ denote the zero-loss set.*

**Definition 4.7** (Distance). *Let $\mathrm{dist}_\mathcal{Z}(\theta) = \inf_{\phi \in \mathcal{Z}} \|\theta - \phi\|$ be the distance from $\theta \in \mathbb{R}^d$ to $\mathcal{Z}$.*

**Theorem 4.8** (Stability of $\mathcal{Z}$). *For every trajectory starting at a zero-loss solution $\theta(0) \in \mathcal{Z}$ and every $\varepsilon > 0$, there exists $\lambda_\varepsilon > 0$ such that for all $0 < \lambda < \lambda_\varepsilon$ the trajectory under $\mathcal{L}_\lambda$ satisfies*

$$\sup_{t \geq 0} \mathrm{dist}_\mathcal{Z}(\theta(t)) < \varepsilon. \tag{4}$$

*Sketch of the Proof.* Our proof is based on the fact that the gradient flow will never increase the optimized quantity $\mathcal{L}_\lambda(\theta) = \mathcal{L}(\theta) + \lambda\|\theta\|^2$. Since both terms are non-negative, we can establish any desired bound on $\mathcal{L}(\theta)$ by an appropriate choice of $\lambda$. We then use this to obtain the bound on the distance. We give the full proof in Appendix A. $\qquad\square$

### 4.4 REGULARITY OF THE ZERO-LOSS SET

We show that the zero-loss set is well-behaved for *almost* every dataset.

**Definition 4.9** (Singular Points). *We say that $\theta \in \mathbb{R}^d$ is a singular point if the Jacobian matrix of $\mathcal{F}$ at $\theta$ is not full rank, i.e. $\mathrm{rank}(\nabla\mathcal{F}(\theta)) < \min(d, km)$. We denote the set of all singular points as $\mathrm{Crit}(\mathcal{F}) \subset \mathbb{R}^{km}$. Note that singular points are defined in terms of $\mathcal{F}$, though they correspond to singular points of $\mathcal{Z}$. With respect to $\mathcal{F}$, they are more precisely* critical points*.*

**Theorem 4.10** (Regularity of $\mathcal{Z}$). *For* almost *every dataset $(x_i, y_i)_{i \leq k}$, the corresponding zero-loss set $\mathcal{Z}$ will not contain any singular points.*

*Sketch of the Proof.* We show that for any set of input vectors $(x_i)_{i \leq k}$, *almost* every possible set of target vectors $(y_i)_{i \leq k}$ induces a zero-loss set $\mathcal{Z}$ that is completely free of singular points.

Using the assumption that $\mathcal{F}$ is smooth, our desired result follows almost immediately from Sard (1942), also known as the *Morse–Sard theorem*, which states that the image of the critical points has Lebesgue measure zero. In our case, the image of the singular points $\mathcal{F}(\mathrm{Crit}(\mathcal{F}))$ has Lebesgue measure zero in the space of possible outputs $\mathbb{R}^{km}$. In other words, only a negligible set of possible outputs is ever hit by a singular point. We give a detailed proof in Appendix B. $\qquad\square$

### 4.5 Loss Gradient Orthogonality

We will now provide our main theoretical result, which states that $\nabla \mathcal{L}(\theta)$ around $\mathcal{Z}$ becomes orthogonal to any tangent direction. We illustrate this concept in Figure 4.

**Definition 4.11** (Tangent Direction). *We say that $v \in \mathbb{R}^d$ is a tangent direction at $\theta \in \mathcal{Z}$ if there exists a smooth trajectory $s : \mathbb{R} \to \mathbb{R}^d$ such that $s(0) = \theta$, $s'(0) = v$, and $\mathcal{L}(s(t)) = 0$ for all $t \in \mathbb{R}$.*

**Definition 4.12** (Tangent Space). *We denote by $T_\theta$ the set of all tangent directions at $\theta \in \mathcal{Z}$.*

**Definition 4.13** (Projection). *Let $\mathrm{proj}_{\mathcal{Z}}(\theta) = \arg\inf_{\theta' \in \mathcal{Z}} \|\theta - \theta'\|$ be the projection of $\theta$ onto $\mathcal{Z}$.*

**Theorem 4.14** (Gradient Orthogonality). *Let $S \subset \mathbb{R}^d$ be a compact space with $\mathrm{proj}_{\mathcal{Z}}(S) \subseteq S$. If $S \cap \mathcal{Z}$ contains no singular points, then there exists a constant $C > 0$ such that*

$$\left| \cos\left( \angle(v, \nabla \mathcal{L}(\theta)) \right) \right| = \left| \frac{v^\top}{\|v\|} \frac{\nabla \mathcal{L}(\theta)}{\|\nabla \mathcal{L}(\theta)\|} \right| < C \operatorname{dist}_{\mathcal{Z}}(\theta) \tag{5}$$

*holds for all $\theta \in S \setminus \mathcal{Z}$ and all tangent directions $v \in T_{\mathrm{proj}_{\mathcal{Z}}(\theta)}$.*

*Sketch of the Proof.* We approximate the loss gradient at $\theta$ using the Taylor expansion around $\mathrm{proj}_{\mathcal{Z}}(\theta)$ to obtain $\nabla \mathcal{L}(\theta) = Hx + O(\|x\|^2)$, where $x = \theta - \mathrm{proj}_{\mathcal{Z}}(\theta)$ and $H = \nabla^2 \mathcal{L}(\mathrm{proj}_{\mathcal{Z}}(\theta))$.

Using $v \in T_{\mathrm{proj}_{\mathcal{Z}}(\theta)}$ and the absence of singular points, we are able to show that $Hv = 0$ and $\|\nabla \mathcal{L}(\theta)\| = \Theta(\|x\|)$. Therefore, the normalized dot product will be $O(\|x\|)$. We give the full proof in Appendix C. $\qquad\square$

**Remark 4.15.** *In other words, the loss gradient does not induce any movement near $\mathcal{Z}$. After a memorizing solution is reached, learning will be driven entirely by weight decay. The loss will only serve to keep the model near $\mathcal{Z}$, while weight decay will be free to push the model towards norm minimization along any of the tangent directions.*

### 4.6 Empirical Validation

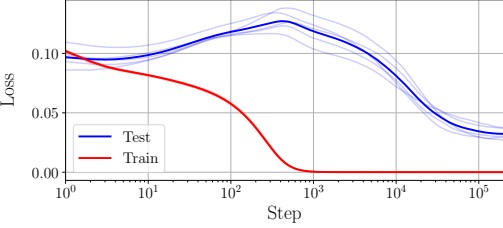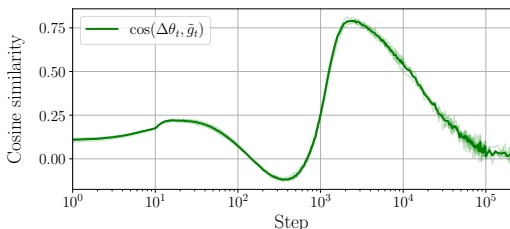

Figure 5: We train smalls networks to perform modular addition. We display their average train and test loss (left) and the average cosine similarity between the parameter updates and the norm-minimizing direction on the zero-loss set (right). As we can see from the loss plot (left), the network exhibits grokking. Interestingly, the similarity is greatest exactly during the *grokking* stage.

We empirically validate our theory that post-memorization dynamics follows the norm-minimization direction on the zero-loss manifold. We train a few small two-layer networks to perform modular addition with ReLU activation, mean squared-error loss, and weight decay. At every step $t$ during training, we measure the cosine similarity between the parameter update $\Delta \theta_t$ and an estimate of the norm-minimizing direction on the zero-loss set $\tilde{g}_t = \arg\min_v(v^\top \theta_t)$ such that $v \in T_{\mathrm{proj}_{\mathcal{Z}}(\theta_t)}$. We display the results in Figure 5. We explain the full experimental details in Appendix D.

## 5 ISOLATED DYNAMICS OF A NETWORK COMPONENT

The parameter vector can be decomposed into two orthogonal parameter subsets $\theta = [\theta_1, \theta_2]$, where $\theta_1 \in \mathbb{R}^{d_1}$, $\theta_2 \in \mathbb{R}^{d_2}$, and $d_1 + d_2 = d$. We are interested in the learning dynamics of $\theta_1$:

$$\dot{\theta}_1 = -\nabla_{\theta_1} \mathcal{L}_\lambda(\theta_1, \theta_2) \tag{6}$$

The gradient flow assumption means that the trajectory of $\theta_1$ is a one-dimensional curve in a $d_1$-dimensional space. This suggests that it is highly unlikely that our trajectory will pass through the same $\theta_1$ twice. If we assume that a training trajectory only goes through unique values of $\theta_1$, then it is possible to parametrize $\theta_2$ as a function of $\theta_1$:

$$\theta_2 = \phi(\theta_1) \tag{7}$$

where $\phi : \mathbb{R}^{d_1} \to \mathbb{R}^{d_2}$ is a function specific to the loss function and the initial parameters.

We motivate this assumption based on the comprehensive literature on multiple points of stochastic processes. For example, even with a dimensionality as small as $d \geq 4$, it is known that a Brownian motion in $\mathbb{R}^d$ contains no self-intersections *almost surely* (Mörters & Peres (2010), Chapter 9). More generally, Dalang et al. (2021) prove the non-existance of multiple points for a wide class of Gaussian random fields.

This parametrization allows us to isolate the dynamics of $\theta_1$ along the training trajectory by expressing them as a function of $\theta_1$ alone:

$$\dot{\theta}_1 = -\nabla_{\theta_1} \mathcal{L}_\lambda(\theta_1, \phi(\theta_1)) \tag{8}$$

While the function $\phi$ is generally intractable, working with reasonable approximations can provide valuable insights into the learning dynamics of $\theta_1$.

### 5.1 APPROXIMATE COST FUNCTION

We propose approximating $\phi$ by assuming that parameters $\theta_2$ are optimal for the current value of $\theta_1$:

$$\phi(\theta_1) = \arg\min_{\theta_2} \mathcal{L}_\lambda(\theta_1, \theta_2) \tag{9}$$

This approximation can also be understood as treating $\theta_1$ as the slow learning component, while $\theta_2$ is the fast learning component that quickly adapts to the current value of $\theta_1$.

Additionally, optimizing $\theta$ under this approximation is equivalent to optimizing the following cost function:

$$\mathcal{R}(\theta_1) = \min_{\theta_2} \mathcal{L}_\lambda(\theta_1, \theta_2) \tag{10}$$

**Theorem 5.1.** *The learning dynamics of $\theta_1$ under Equations* (8) *and* (9) *follow the gradient flow of $\mathcal{R}$ when $\phi$ is differentiable:*

$$\dot{\theta}_1 = -\nabla_{\theta_1} \mathcal{R}(\theta_1) \tag{11}$$

*Proof.* Note that $\mathcal{R}(\theta_1) = \mathcal{L}_\lambda(\theta_1, \phi(\theta_1))$. By differentiating it with respect to $\theta_1$ we obtain that $\nabla_{\theta_1} \mathcal{R}(\theta_1) = \nabla_{\theta_1} \mathcal{L}_\lambda(\theta_1, \phi(\theta_1)) + \nabla_{\theta_2} \mathcal{L}_\lambda(\theta_1, \phi(\theta_1)) \nabla_{\theta_1} \phi(\theta_1)$. However, since $\phi(\theta_1)$ is a minimum of $\mathcal{L}_\lambda$, we have that $\nabla_{\theta_2} \mathcal{L}_\lambda(\theta_1, \phi(\theta_1)) = 0$, giving us the desired result. ∎

## 6 TWO-LAYER NETWORKS

### 6.1 SETUP

We turn our attention to the learning dynamics of a two-layer neural network trained with mean squared error loss and weight decay:

$$\mathcal{L} = \| \sigma(X W_1) W_2 - Y \|_F^2 \tag{12}$$

where $X \in \mathbb{R}^{n \times d_{in}}$ is the input data, $Y \in \mathbb{R}^{n \times d_{out}}$ is the target output, $\sigma : \mathbb{R} \to \mathbb{R}$ is the activation function, $W_1 \in \mathbb{R}^{d_{in} \times d_h}$ is the first layer weights, $W_2 \in \mathbb{R}^{d_h \times d_{out}}$ is the second layer weights, and $\|\cdot\|_F$ denotes the Frobenius norm.

After applying a weight decay coefficient $\lambda > 0$, we get:

$$\mathcal{L}_\lambda = \mathcal{L} + \lambda \left( \|W_1\|_F^2 + \|W_2\|_F^2 \right) \tag{13}$$

## 6.2 Isolated Dynamics of the First Layer

Using the approximation from Section 5.1, we can isolate the learning dynamics of the first layer by assuming that the second layer weights are optimal for the current value of the first layer weights:

$$W_2 \approx \phi(W_1) = \arg \min_{\tilde{W}_2} \mathcal{L}_\lambda(W_1, \tilde{W}_2) \tag{14}$$

Since the second layer is just a linear transformation of the hidden layer activations $H = \sigma(XW_1)$, finding the optimal second layer weights is equivalent to the classic problem of ridge regression (Hoerl & Kennard, 1970). The solution is given by:

$$\phi(W_1) = (H^\top H + \lambda I)^{-1} H^\top Y \tag{15}$$

By combining Equations (12), (13) and (15), we can obtain the cost function for the isolated learning dynamics of the first layer:

$$\mathcal{R}(W_1) = \mathcal{L}_\lambda(W_1, \phi(W_1)) \tag{16}$$

This cost function is not particulary simple, but it is fully differentiable, allowing us to approximate the learning dynamics of the first layer:

$$\dot{W}_1 \approx -\nabla \mathcal{R}(W_1) \tag{17}$$

## 6.3 Zero-Loss Approximation

We further assume a strongly overparametrized regime with more hidden units that training samples ($d_h > n$). This allows the second layer to fit the outputs perfectly almost always. Following the theoretical framework developed in Section 4, we can further simplify equation Equation (15) by working in the limit of very small weight decay $\lambda \to 0$:

$$\phi(W_1) = H^+ Y \tag{18}$$

where $H^+$ is the Moore-Penrose pseudo-inverse of $H$. When $H$ has linearly independent columns, then the pseudo-inverse is given by $H^+ = (H^\top H)^{-1} H^\top$. If $H$ has linearly independent rows, then $H^+ = H^\top (HH^\top)^{-1}$. Note that $H \in \mathbb{R}^{n \times d_h}$. Given the overparameterized regime ($d_h > n$), we use the latter. This allows us to further simplify the cost function down to:

$$\mathcal{R}(W_1) = \lambda \|W_1\|_F^2 + \lambda \operatorname{Tr}\left(Y^\top (HH^\top)^{-1} Y\right) \tag{19}$$

By differentiating this cost function, we can obtain a closed-form expression for the isolated learning dynamics of the first layer in the overparameterized zero-loss approximation:

$$\dot{W}_1 \approx X^\top \left( \left(AYY^\top AH\right) \odot \sigma'\left(XW_1\right) \right) - W_1 \tag{20}$$

where $H = \sigma(XW_1)$, $A = (HH^\top)^{-1}$, $\sigma$ is the activation function, and $\odot$ denotes the Hadamard product. We provide a detailed derivation in Appendix E.

## 7 Simulated Dynamics

In this section, we empirically validate our combined theoretical insights on isolated dynamics and post-memorization dynamics. By applying equation Equation (20) to a network trained on the modular addition task, we show that our approximations reproduce the delayed generalization and circular representations characteristic of grokking.

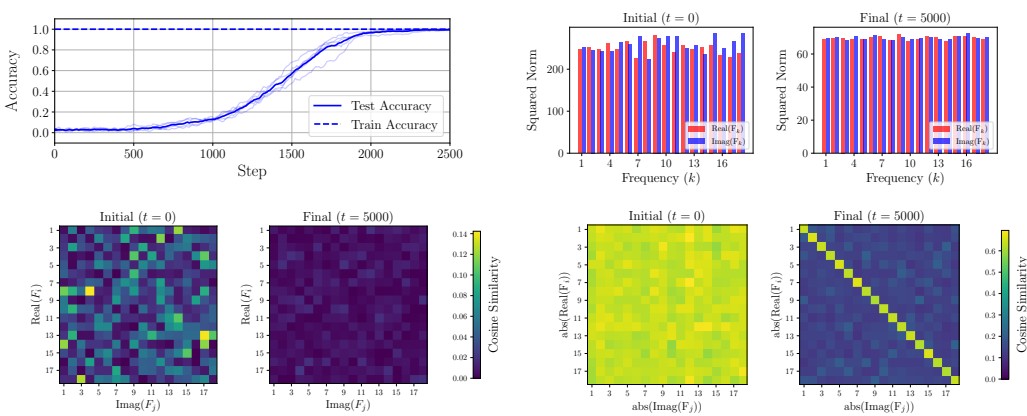

Figure 6: Simulated dynamics according to Equation (20) reproduce the phenomena of delayed generalization and representation learning. *Top left:* generalization emerges after about 1000 steps, despite training loss being exactly zero throughout. *Top right:* Fourier features norms equalize, suggesting the presence of equally-sized circles. *Bottom left:* Fourier features become orthogonal, suggesting that circles are located in orthogonal planes. *Bottom right:* Fourier features absolute values become dissimilar, suggesting that each circle leverages a different subset of hidden activations.

## 7.1 DATASET

We train the network to perform modular addition modulo a fixed number $p$. The dataset consists of $k = p(p+1)/2$ unique input pairs and their sum, $D = \{(a, b, c) \mid 0 \leq a \leq b < p, \ c = (a+b) \bmod p\}$. We construct the input data $X \in \mathbb{R}^{k \times p}$ and the target output $Y \in \mathbb{R}^{k \times p}$ as $X_i = e_{a_i} + e_{b_i}$ and $Y_i = e_{c_i}$ for all $i = 1, \ldots, k$, where $e_i$ is the $i$-th unit vector in $\mathbb{R}^p$ and $(a_i, b_i, c_i) \in D$ is the $i$-th sample in the dataset.

We split the dataset into $(X_{\text{train}}, Y_{\text{train}})$ and $(X_{\text{test}}, Y_{\text{test}})$, using a fraction $f_s$ of the dataset for training and the remaining $1 - f_s$ for testing.

## 7.2 ARCHITECTURE

We train a two-layer neural network with input dimension $p$, hidden dimension $d_h$, output dimension $p$, and a non-linear activation $\sigma : \mathbb{R} \to \mathbb{R}$.

Since the inputs are sums of one-hot vectors, we refer to the first layer weights as the embedding matrix $E \in \mathbb{R}^{p \times d_h}$. We refer to the second layer weights as simply the weights matrix $W \in \mathbb{R}^{d_h \times p}$.

The network output is given by $\hat{Y} = \sigma(XE)W$. To emphasize the role of the first layer as an embedding, we can also write the output as $\hat{Y}_i = \sigma(E_{a_i} + E_{b_i})W$.

We define the test accuracy as the percentage of correctly predicted test samples. We say that a test sample is correctly predicted if the index of the maximum value in the predicted output $\hat{Y}_i$ matches $c_i$.

## 7.3 SIMULATED OPTIMIZATION

Our goal is not to train the network, but to validate that our approximate learning dynamics reproduce the phenomena observed during standard training.

We simulate the evolution of the embedding matrix $E$ under the isolated dynamics given by equation Equation (20). We start from a random initialization $E \sim \mathcal{N}(0, \ p^{-1/2})$ and update it for $T$ steps as $E \leftarrow E + \eta \Delta E$, where $\eta > 0$ is the step size and $\Delta E = X^\top \left( \left( AYY^\top AH \right) \odot \sigma'(XE) \right) - E$.

The isolated dynamics assume that $W = (H^\top H)^{-1} H^\top Y$ is optimal for the current value of $E$, which guarantees zero loss and perfect accuracy on the training data throughout training. This also ensures that predicted outputs perfectly match the target outputs on the training data, which is less principled for a classification task, but generally performs well in practice (Rifkin et al., 2003).

**Details.** We use $p = 37$, $d_h = 512$, $\sigma(x) = \max(0, x)$, $f_s = 0.7$, $\eta = 10^{-3}$, $T = 5000$.

### 7.3.1 DELAYED GENERALIZATION

We simulate 5 runs starting from different random initializations and plot the test accuracy in Figure 6. Despite the fact that the training loss is exactly zero throughout, the test accuracy is not better than random guessing for the first 500 steps. However, the network eventually achieves perfect generalization on the test data after about 2000 steps, reproducing the delayed generalization phenomena Power et al. (2022).

### 7.3.2 FOURIER FEATURES

Using a discrete Fourier transform, we decompose the embedding matrix $E$ into a linear combination of circles with different frequencies:

$$F_k = \frac{1}{p} \sum_{j=0}^{p-1} e^{-i2\pi\, jk/p}\, E_j \qquad \forall k \in \{1, \ldots, (p-1)/2\}$$

Projecting the embeddings onto the plane spanned by $\mathrm{Re}(F_k)$ and $\mathrm{Im}(F_k)$ gives us a circle where the embeddings appear in the order $\{0,\, k,\, 2k,\, 3k,\, \ldots,\, (p-1)k\} \bmod p$. Note that a circle of frequency $k$ is equivalent to a circle of frequency $p-k$, so we only need to consider frequencies up to $(p-1)/2$.

We visualize several comparisons of the Fourier features of the initial and final embedding matrices for a single run Figure 6. First, the norms of the real and imaginary parts of the Fourier features equalize, suggesting the presence of *equally sized circles with perfect aspect ratios*. Second, the real and imaginary parts of the Fourier features become orthogonal, indicating that *circles are located in orthogonal planes*. Third, by taking the absolute value of the real and imaginary parts of the Fourier features, we obtain vectors very similar for the same frequency, but very different for different frequencies. This suggests that *each circle leverages a different subset of hidden units*.

## 8 CONCLUSION

We have formally established that the learning dynamics of neural networks in the grokking regime approximate as the minimization of the weight norm within the zero-loss set. Additionally, we have established a theoretical basis for approximating the learning dynamics of individual network components.

**Limitations.** This work does not cover cross-entropy loss, which is commonly used in practice. With regard to isolated dynamics, our work is limited to the case of two-layer networks. Exciting challenges lie ahead in understanding the grokking dynamics of more complex settings and architectures.

**Impact Statement.** We believe that understanding the learning dynamics of neural networks is essential for the design of more efficient and accurate AI systems. However, the development and deployment of such systems should be approached with caution.

**LLM Usage.** Large Language Models (LLMs) were used in standard ways throughout this work to polish the writing, assist with coding, and support brainstorming of mathematical proofs.

**Reproducibility.** We provide the full code for training and plotting used for the experiments from Sections 4.6 and 7.

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

# A    PROOF OF THEOREM 4.8 (STABILITY OF $\mathcal{Z}$)

We begin by establishing the following intermediate result:

**Lemma A.1.** *For every initialization $\theta(0) \in \mathcal{Z}$ and every $\varepsilon > 0$, there exists $\lambda_\varepsilon > 0$ such that for all $0 < \lambda < \lambda_\varepsilon$ the corresponding trajectory $\theta(t)$ under $\mathcal{L}_\lambda$ satisfies*

$$\sup_{t \geq 0} \mathcal{L}(\theta(t)) < \varepsilon. \tag{21}$$

*Proof.* From Equation (3), we see that gradient flow will never increase the optimized quantity $\mathcal{L}_\lambda(\theta) = \mathcal{L}(\theta) + \lambda\|\theta\|^2$. By choosing $\lambda_\varepsilon < \varepsilon/\|\theta(0)\|^2$, we ensure that $\mathcal{L}(\theta(t)) < \varepsilon$ for all $t \geq 0$. □

We recall Theorem 4.8:

**Theorem 4.8** (Stability of $\mathcal{Z}$). *For every trajectory starting at a zero-loss solution $\theta(0) \in \mathcal{Z}$ and every $\varepsilon > 0$, there exists $\lambda_\varepsilon > 0$ such that for all $0 < \lambda < \lambda_\varepsilon$ the trajectory under $\mathcal{L}_\lambda$ satisfies*

$$\sup_{t \geq 0} \mathrm{dist}_{\mathcal{Z}}(\theta(t)) < \varepsilon. \tag{4}$$

*Proof.* Our training trajectory will not reach any $\theta \in \mathbb{R}^d$ with $\|\theta\| > \|\theta(0)\|$. This is because any such configuration is unreachable by gradient flow from $\theta(0)$ for any $\lambda > 0$ since $\mathcal{L}_\lambda(\theta) > \mathcal{L}_\lambda(\theta_0)$.

We are left to show unreachability of the set $\Phi = \{\theta \in \mathbb{R}^d : \mathcal{D}(\theta) \geq \varepsilon \text{ and } \|\theta\| \leq \|\theta(0)\|\}$. Since $\Phi$ is compact, $m = \min_\Phi \mathcal{L}(\theta)$ exists and is positive. Applying Lemma A.1, there exists $\lambda > 0$ such that optimizing $\mathcal{L}_\lambda$ starting from $\theta(0)$ is guaranteed to maintain $\mathcal{L}(\theta(t)) < m$, thus making $\Phi$ unreachable. □

# B    PROOF OF THEOREM 4.10 (REGULARITY OF $\mathcal{Z}$)

We consider the inputs vectors $x_i \in \mathbb{R}^n$ fixed and we show that singularities will not be encountered for *almost all* possible target outputs $y_i \in \mathbb{R}^m$, where $i \in \{1, \ldots, k\}$. Recall that we are training on $k$ samples with input dimension $n$ and output dimension $m$.

We denote the concatenated network outputs with network parameters $\theta \in \mathbb{R}^d$ using $\mathcal{F} : \mathbb{R}^d \to \mathbb{R}^{km}$, where

$$\mathcal{F}(\theta) = \left[ \ f(\theta, x_1)^\top, \ f(\theta, x_2)^\top, \ \ldots, f(\theta, x_n)^\top \ \right]^\top. \tag{22}$$

We denote the concatenated target outputs as $y = \left[ \ y_1^\top, \ y_2^\top, \ \ldots, \ y_k^\top \ \right]^\top \in \mathbb{R}^{km}$.

For any vector of target outputs $y \in \mathbb{R}^{km}$, we denote the set of parameters that fit them as

$$\mathcal{Z}_y = \left\{ \ \theta \in \mathbb{R}^d \ \middle| \ \mathcal{F}(\theta) = y \ \right\}. \tag{23}$$

Note that $\mathcal{Z}_y$ is exactly the zero-loss set with target outputs $y \in \mathbb{R}^{km}$.

We will show that $\mathcal{Z}_y$ contains no singular points for *almost* every $y \in \mathbb{R}^{km}$. More precisely, we show that $\mathcal{Z}_y \cap \mathrm{Crit}(\mathcal{F}) = \emptyset$ for all $y \in \mathbb{R}^{km}$ except a set of Lebesgue measure zero.

We denote the set of target output vectors that lead to singularities as

$$\xi \ = \ \left\{ \ y \in \mathbb{R}^{km} \ \middle| \ \mathcal{Z}_y \cap \mathrm{Crit}(\mathcal{F}) \neq \emptyset \ \right\} \tag{24}$$

**Proposition B.1.** *The set of target outputs that lead to singularities is exactly the image of the set of singular points:*

$$\xi \ = \ \mathcal{F}\big(\mathrm{Crit}(\mathcal{F})\big). \tag{25}$$

*Proof.* By our definitions, we have that $y \in \xi$ if and only if there exists $\theta \in \mathbb{R}^d$ such that $\mathcal{F}(\theta) = y$ and $\theta \in \mathrm{Crit}(\mathcal{F})$. First, any $\theta \in \mathrm{Crit}(\mathcal{F})$ implies that $\mathcal{F}(\theta) \in \xi$. Second, for some $y \in \mathbb{R}^{km}$, if no $\theta \in \mathrm{Crit}(\mathcal{F})$ exists such that $\mathcal{F}(\theta) = y$, then $y \notin \xi$. $\qquad\square$

**Theorem B.2** (Morse–Sard theorem). *If a function $g : \mathbb{R}^a \to \mathbb{R}^b$ is continuously differentiable $k$ times, where $k \geq \max(a - b + 1, 1)$, then the image of its critical set $g\big(\mathrm{Crit}(g)\big)$ has Lebesgue measure zero in $\mathbb{R}^b$.*

This result was first proved by Morse (1939) for the single-output functions (i.e., $b = 1$), and later generalized to differentiable maps by Sard (1942).

By Assumption 4.2, the function $\mathcal{F}$ fits the continuous differentiability criteria. Therefore, we get that the set of target vectors that lead to singularities $\xi = \mathcal{F}\big(\mathrm{Crit}(\mathcal{F})\big) \subset \mathbb{R}^{km}$ has Lebesgue measure zero in $\mathbb{R}^{km}$.

## C  PROOF OF THEOREM 4.14 (GRADIENT ORTHOGONALITY)

We begin with a few intermediate results.

**Lemma C.1.** *At a non-singular point of the zero-loss set, the tangent space is exactly the null space of the Hessian matrix, i.e. $T_\theta = \{ v \in \mathbb{R}^d \mid \nabla^2 \mathcal{L}(\theta)\, v = 0 \}$ for any non-singular $\theta \in \mathcal{Z}$.*

*Proof.* Part I: $\nabla^2 \mathcal{L}(\theta)\, v = 0 \implies v \in T_\theta$.

We write the loss function as

$$\mathcal{L}(\theta) = \|\mathcal{F}(\theta) - y_{all}\|^2$$

using $y_{all} \in \mathbb{R}^{km}$ as the concatenation of all target outputs:

$$y_{all} = \begin{bmatrix} y_1^\top, \ y_2^\top, \ \ldots, \ y_k^\top \end{bmatrix}^\top$$

Differentiating $\mathcal{L}(\theta)$ with respect to $\theta$, we obtain the gradient:

$$\nabla \mathcal{L}(\theta) = 2 \nabla \mathcal{F}(\theta)^\top \big(\mathcal{F}(\theta) - y\big),$$

where $\nabla \mathcal{F}(\theta) \in \mathbb{R}^{km \times d}$ is the Jacobian of the network output with respect to the parameters.

Differentiating again, we obtain the Hessian:

$$\nabla^2 \mathcal{L}(\theta) = 2 \nabla \mathcal{F}(\theta)^\top \nabla \mathcal{F}(\theta) + 2 \sum_{i=1}^{km} \big(\mathcal{F}_i(\theta) - (y_{all})_i\big) \nabla^2 \mathcal{F}_i(\theta),$$

where $\nabla^2 \mathcal{F}_i(\theta) \in \mathbb{R}^{d \times d}$ is the Hessian of the $i$-th output component.

At $\theta \in \mathcal{Z}$, where $\mathcal{F}(\theta) = y_{all}$, the second term vanishes, simplifying the Hessian to:

$$\nabla^2 \mathcal{L}(\theta) = 2 \nabla \mathcal{F}(\theta)^\top \nabla \mathcal{F}(\theta).$$

For any direction $v \in \mathbb{R}^d$, this yields:

$$v^\top \nabla^2 \mathcal{L}(\theta) v = 2 \|\nabla \mathcal{F}(\theta) v\|^2.$$

Therefore,

$$v^\top \nabla^2 \mathcal{L}(\theta)\, v = 0 \quad \Leftrightarrow \quad \nabla \mathcal{F}(\theta)\, v = 0.$$

Moreover, every $\theta \in \mathcal{Z}$ is a local minimum where the Hessian matrix is symmetric and positive semi-definite. This gives the us equivalence

$$\nabla^2 \mathcal{L}(\theta)\, v = 0 \quad \Leftrightarrow \quad v^\top \nabla^2 \mathcal{L}(\theta)\, v = 0.$$

Therefore,

$$\nabla^2 \mathcal{L}(\theta)\, v = 0 \quad \Leftrightarrow \quad \nabla \mathcal{F}(\theta)\, v = 0.$$

Because $km \geq d$ and $\nabla \mathcal{F}(\theta)$ has full rank, the *inverse function theorem* implies that the preimage of $\mathcal{F}$ locally has the structure of a smooth manifold whose tangent space is exactly the null space of $\nabla \mathcal{F}(\theta)$. To simplify our analysis, we directly restate the inverse function theorem below in a form that is slightly non-standard, but perfectly equivalent:

**Theorem C.2** (Inverse Function Theorem). *Assume that $\mathcal{F} : \mathbb{R}^d \to \mathbb{R}^{km}$ is a continuously differentiable function with $d \geq km$ and $\mathrm{rank}(\nabla \mathcal{F}(\theta)) = km$ for some $\theta \in \mathbb{R}^d$. Then, for all $v \in R^d$ such that $\nabla \mathcal{F}(\theta)\, v = 0$, there exists a smooth trajectory $s : R \to R^d$ such that $s(0) = \theta$, $s'(0) = v$, and $\mathcal{F}(s(t)) = \mathcal{F}(\theta)$ for all $t \in \mathbb{R}$.*

Note that any such trajectory will also have $\mathcal{L}(s(t)) = 0$ since $\mathcal{F}(s(t)) = \mathcal{F}(\theta) = y_{all}$. This implies the desired result.

*Part II:* $v \in T_\theta \implies \nabla^2 \mathcal{L}(\theta) \, v = 0.$

Using the equation $\mathcal{L}(f(t)) = 0$ from Definition 4.11 and differentiating it twice with respect to $t$, we obtain:

$$f'(t)^\top \nabla^2 \mathcal{L}(f(t)) f'(t) + \nabla \mathcal{L}(f(t))^\top f''(t) = 0. \tag{26}$$

Since any point $\theta \in \mathcal{Z}$ is a local minimum, we have that $\nabla \mathcal{L}(\theta) = 0$ and $\nabla^2 \mathcal{L}(\theta)$ is symmetric and positive semi-definite (PSD).

By evaluating Equation (26) at $t = 0$, we obtain that $v^\top \nabla^2 \mathcal{L}(\theta) \, v = 0$. Since $\nabla^2 \mathcal{L}(\theta)$ is symmetric and PSD, this implies the desired result.

$\square$

**Definition C.3** (Normal Space). *Let* $\mathcal{N}_\phi = \{ \, \alpha \in \mathbb{R}^d \mid \alpha^\top v = 0, \ \forall v \in T_\phi \, \}$ *denote the normal space at a point* $\phi \in \mathcal{Z}$.

**Proposition C.4.** *The displacement of a point from the zero-loss set belongs to the normal space at the point's projection, i.e.* $\theta - \mathrm{proj}_{\mathcal{Z}}(\theta) \in \mathcal{N}_{\mathrm{proj}_{\mathcal{Z}}(\theta)}$ *for all* $\theta \in \mathbb{R}^d$.

*Proof.* We give a proof by contradiction. Assume that there exists $v \in T_{\mathrm{proj}_{\mathcal{Z}}(\theta)}$ such that $v^\top (\theta - \mathrm{proj}_{\mathcal{Z}}(\theta)) \neq 0$. Then, by Definition 4.11, there must exist a smooth trajectory $f : \mathbb{R} \to \mathbb{R}^d$ with $f(0) = \mathrm{proj}_{\mathcal{Z}}(\theta)$, $f'(0) = v$, and $f(t) \in \mathcal{Z}$ for all $t \in R$. By moving $\mathrm{proj}_{\mathcal{Z}}(\theta)$ along this trajectory, we can get closer to $\theta$. However, this should not be possible according to Definition 4.13.

$\square$

We now establish our main result:

**Theorem 4.14** (Gradient Orthogonality). *Let* $S \subset \mathbb{R}^d$ *be a compact space with* $\mathrm{proj}_{\mathcal{Z}}(S) \subseteq S$. *If* $S \cap \mathcal{Z}$ *contains no singular points, then there exists a constant* $C > 0$ *such that*

$$\left| \cos \left( \angle (v, \nabla \mathcal{L}(\theta)) \right) \right| = \left| \frac{v^\top}{\|v\|} \frac{\nabla \mathcal{L}(\theta)}{\|\nabla \mathcal{L}(\theta)\|} \right| < C \operatorname{dist}_{\mathcal{Z}}(\theta) \tag{5}$$

*holds for all* $\theta \in S \setminus \mathcal{Z}$ *and all tangent directions* $v \in T_{\mathrm{proj}_{\mathcal{Z}}(\theta)}$.

*Proof.* By parameterizing $\theta$ as

$$\theta = \mathrm{proj}_{\mathcal{Z}}(\theta) + \|\theta - \mathrm{proj}_{\mathcal{Z}}(\theta)\| \frac{\theta - \mathrm{proj}_{\mathcal{Z}}(\theta)}{\|\theta - \mathrm{proj}_{\mathcal{Z}}(\theta)\|}$$

we can denote the quantity of interest as

$$\frac{v^\top \nabla \mathcal{L}(\theta)}{\|v\| \, \|\nabla \mathcal{L}(\theta)\|} = g \left( \mathrm{proj}_{\mathcal{Z}}(\theta), \ \frac{\theta - \mathrm{proj}_{\mathcal{Z}}(\theta)}{\|\theta - \mathrm{proj}_{\mathcal{Z}}(\theta)\|}, \ \|\theta - \mathrm{proj}_{\mathcal{Z}}(\theta)\|, \ v \right)$$

where

$$g(\phi, \alpha, x, v) = \frac{v^\top \nabla \mathcal{L}(\phi + x\alpha)}{\|v\| \, \|\nabla \mathcal{L}(\phi + x\alpha)\|}$$

with $\phi \in \mathcal{Z}$, $\alpha \in \mathcal{N}_\phi$, $x \in R^+$, and $v \in T_\phi$.

To obtain the desired result, it suffices to show that there exists $C > 0$ such that

$$g(\phi, \alpha, x, v) < Cx$$

for all $\phi \in S \cap \mathcal{Z}$, $\alpha \in U(\mathcal{N}_\phi)$, $x > 0$, $v \in T_\phi$, where $U(\mathcal{N}_\phi) = \{ \, v \in \mathcal{N}_\phi \mid \|v\| = 1 \, \}$.

We write the gradient of the loss function around $\phi \in \mathcal{Z}$ using the Taylor expansion:

$$\nabla \mathcal{L}(\phi + x\alpha) = x \, \nabla^2 \mathcal{L}(\phi) \, \alpha + x \, h_{\phi, \alpha}(x)$$

where $h_{\phi, \alpha}(x)$ is a remainder term that vanishes as $x \to 0$. In other words, there exist constants $M_{\phi, \alpha}, \, a_{\phi, \alpha} > 0$ such that

$$\|h_{\phi, \alpha}(x)\| < M_{\phi, \alpha} \, x \qquad \forall x, \, x < a_{\phi, \alpha}.$$

Since $S$ is closed and $\mathcal{L}$ is continuous, $S \cap \mathcal{Z}$ will also be closed. The set of normal vectors with unit norm at any point is also closed. This implies that the following quantities exist and are positive:

$$M_{\text{sup}} = \sup_{\substack{\phi \in S \cap \mathcal{Z} \\ \alpha \in U(\mathcal{N}_\psi)}} M_{\phi, \alpha} \qquad\qquad a_{\text{inf}} = \inf_{\substack{\phi \in S \cap \mathcal{Z} \\ \alpha \in U(\mathcal{N}_\psi)}} a_{\phi, \alpha}$$

We express $g(\theta, \alpha, x, v)$ as:

$$g(\phi, \alpha, x, v) = \frac{x\, v^\top \nabla^2 \mathcal{L}(\phi)\, \alpha + x\, v^\top h_{\phi, \alpha}(x)}{\|v\|\, \|x\, \nabla^2 \mathcal{L}(\phi)\, \alpha + x\, h_{\phi, \alpha}(x)\|} = \frac{v^\top \nabla^2 \mathcal{L}(\phi)\, \alpha + v^\top h_{\phi, \alpha}(x)}{\|v\|\, \|\nabla^2 \mathcal{L}(\phi)\, \alpha + h_{\phi, \alpha}(x)\|}$$

Since $v \in T_\phi$, from Lemma C.1, we have that $v^\top \nabla^2 \mathcal{L}(\phi)\, \alpha = 0$. This gives

$$g(\phi, \alpha, x, v) = \frac{v^\top h_{\phi, \alpha}(x)}{\|v\|\, \|\nabla^2 \mathcal{L}(\phi)\, \alpha + h_{\phi, \alpha}(x)\|}$$

$$\leq \frac{\|h_{\phi, \alpha}(x)\|}{\|\nabla^2 \mathcal{L}(\phi)\, \alpha + h_{\phi, \alpha}(x)\|}$$

Since $\nabla^2 \mathcal{L}(\phi)\, \alpha \neq 0$, the following also exists and is positive:

$$\lambda_{\text{inf}} = \inf_{\substack{\phi \in S \cap \mathcal{Z} \\ \alpha \in U(\mathcal{N}_\psi)}} \|\nabla^2 \mathcal{L}(\phi)\, \alpha\|$$

Assuming that $x < \lambda_{\text{inf}} / M_{\text{sup}}$ guarantees that $\|\nabla^2 \mathcal{L}(\phi)\, \alpha\| > \|h_{\phi, \alpha}(x)\|$, which gives

$$g(\phi, \alpha, x, v) \leq \frac{\|h_{\phi, \alpha}(x)\|}{\|\nabla^2 \mathcal{L}(\phi)\, \alpha\| - \|h_{\phi, \alpha}(x)\|}$$

$$\leq \frac{M_{\text{sup}}\, x}{\lambda_{\text{inf}} - M_{\text{sup}}\, x}$$

for any $x < a_{\text{inf}}$. An appropriate choice of $C$ gives the desired bound for all $\theta$ with $\text{dist}_{\mathcal{Z}} \theta < x_0$ for some $x_0 > 0$, for example:

$$x_0 = \min\left(a_{\text{inf}}, \frac{\lambda_{\text{inf}}}{2\, M_{\text{sup}}}\right) \qquad C = \frac{2\, M_{\text{sup}}}{\lambda_{\text{inf}}}$$

Since the LHS of Theorem 4.14 is bounded by 1, points with $\text{dist}_{\mathcal{Z}} \theta > x_0$ will always satisfy the bound with $C = 1/x_0$. Therefore, we can absorb $x_0$ into $C$ as

$$C = \max\left(\frac{1}{x_0}, \frac{2\, M_{\text{sup}}}{\lambda_{\text{inf}}}\right). \tag{27}$$

$\square$

## D  EXPERIMENTAL DETAILS FOR SECTION 4.6 (SIMILARITY OF DYNAMICS)

We empirically validate our theory that post-memorization dynamics follows the norm-minimization direction on the zero-loss manifold. We train a few small networks to perform modular addition using two layer, ReLU activation, mean squared-error loss, and weight decay.

**Setup.**  We use the same architecture and dataset as in Section 7. We train two-layer networks to perform modular addition with two numbers with a fixed modulo $n$. The network is given two numbers $a, b \in \{1, \ldots, N\}$ and must output their sum. The network has $n$ input neurons, $h$ hidden neurons, and $n$ output neurons. ReLU activation is applied on the hidden layer. The input is the sum of the one-hot vector representations of $a$ and $b$. The target output is the one-hot vector representation of $a + b \mod n$. See Section 7 for more details.

**Training Details.**  In order to accelerate the calculation of the Jacobian matrix, we use very small networks with $n = 11$ and $h = 128$. There are $n(n + 1)/2 = 77$ pairs of numbers in the data set. We use 7 of them as a test set and the others as training data. We train five networks with full-batch gradient descent learning rate $\eta = 1$, weight decay $\lambda = 10^{-4}$, and no momentum. We don't use biases and we initialize weight matrices using the standard PyTroch initialization.

**Parameter Update Estimation.**  We estimate the parameter update as $\Delta\theta_n = \theta_n - \theta_{\max(0, n-c)}$ with $c = 10$. This averaging over $c$ update steps gives a better estimation of the direction of the training trajectory by canceling the noisy high-frequency oscillations that might be contained in a single update step.

**Norm-Minimizing Direction Estimation.**  We aim to estimate the direction that minimizes the norm constrained to the zero-loss set around $\theta_t$. Formally, we are interested in

$$\tilde{g}_t = \arg\min_v (v^\top \theta_t) \qquad \text{s.t. } v \in T_{\text{proj}_{\mathcal{Z}}(\theta_t)} \text{ and } \|v\| = 1 \tag{28}$$

where $T_{\text{proj}_{\mathcal{Z}}(\theta_t)}$ is the tangent space of the zero-loss manifold $\mathcal{Z}$ at the closest point to $\theta_t$. In order to estimate this, first we must estimate the closest zero-loss point to $\theta_t$, namely $\text{proj}_{\mathcal{Z}}(\theta_t)$. We estimate this using 100 steps of gradient descent from $\theta_t$ without weight decay $\lambda = 0$, same learning rate $\eta = 1$, and momentum $\beta = 0.9$ to accelerate convergence. This results in a new point $\theta'_t$ with very low loss. We assume that this point is on the zero loss and we project the parameters on its level set. We achieve this by computing the Jacobian matrix of $\mathcal{F}$ at $\theta'$ and projecting onto its null space:

$$\tilde{g}_t \approx (I - J^+ J)(-\theta_t) \tag{29}$$

where $J = \nabla\mathcal{L}(\theta_t) \in \mathbb{R}^{km \times d}$ is the Jacobian matrix at $\theta'$ and $J^+ = (J^\top J)^{-1} J^\top$ is the Moore–Penrose pseudo-inverse.

**Plotting.**  We train 5 different networks from random initializations and we plot their statistics in Figure 5. We plot the statistics of individual networks using transparent lines, and we plot the averaged statistics per training step using opaque lines. In Figure 5, we plot the loss rather than the percentage accuracy, since the loss provides a smoother measure of progress in our setup with very few test samples, but we also plot the accuracies in Figure 7 for completeness.

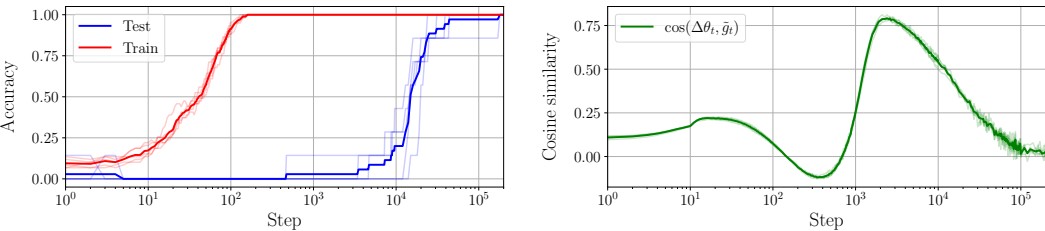

Figure 7: Same as Figure 5, but with percentage accuracy rather than loss plotted in the left figure.

# E COST FUNCTION GRADIENT IN TWO-LAYER NETWORKS

We want to derive the gradient of the following cost function:

$$\mathcal{R}(W_1) = \lambda \|W_1\|_F^2 + \lambda \|\phi(W_1)\|_F^2$$

where

$$\phi(W_1) = H^\top (HH^\top)^{-1} Y \qquad H = \sigma(XW_1) \qquad X \in \mathbb{R}^{n \times d_{\text{in}}} \qquad W_1 \in \mathbb{R}^{d_{\text{in}} \times d_h} \qquad Y \in \mathbb{R}^{n \times d_{\text{out}}}$$

The activation function $\sigma : \mathbb{R} \to \mathbb{R}$ is applied is applied elementwise. Note that, since we work in the zero-loss approximation, there is no explicit error term.

We decompose $\mathcal{R}(W_1)$ into two terms:

$$\mathcal{R}(W_1) = \underbrace{\lambda \|W_1\|_F^2}_{\text{Term 1}} + \underbrace{\lambda \|\phi(W_1)\|_F^2}_{\text{Term 2}}$$

Since the first term is a standard Frobenius norm squared, its gradient is:

$$\nabla_{W_1} \left[ \lambda \|W_1\|_F^2 \right] = 2\lambda W_1$$

To find the gradient of the second term, we analyze:

$$f(W_1) = \|\phi(W_1)\|_F^2 = \|H^\top (HH^\top)^{-1} Y\|_F^2$$

Since the Frobenius norm squared satisfies $\|M\|_F^2 = \text{Tr}(M^\top M)$, we write:

$$f(W_1) = \text{Tr}\left( \left( H^\top (HH^\top)^{-1} Y \right)^\top H^\top (HH^\top)^{-1} Y \right)$$
$$= \text{Tr}\left( Y^\top (HH^\top)^{-1} HH^\top (HH^\top)^{-1} Y \right)$$
$$= \text{Tr}\left( YY^\top (HH^\top)^{-1} \right)$$

Using the following known result from matrix calculus:

$$\frac{\partial \text{Tr}(MP^{-1})}{\partial P} = -P^{-1} M P^{-1}$$

with $M = YY^\top$ and $P = HH^\top$, we obtain:

$$\frac{\partial f}{\partial H} = -2(HH^\top)^{-1} YY^\top (HH^\top)^{-1} H$$

Propagating the gradient using the chain rule, we get:

$$\frac{\partial f}{\partial W_1} = -2X^\top \left[ \left( (HH^\top)^{-1} YY^\top (HH^\top)^{-1} H \right) \odot \sigma'(XW_1) \right]$$

where $\odot$ denotes the Hadamard product.

Finally, multiplying by $\lambda$, we obtain the gradient of the second term:

$$\nabla_{W_1} \left[ \lambda \|\phi(W_1)\|_F^2 \right] = -2\lambda X^\top \left[ (AYY^\top AH) \odot \sigma'(XW_1) \right]$$

where $A = (HH^\top)^{-1}$.

Thus, the final expression is:

$$\nabla \mathcal{R}(W_1) = -2\lambda W_1 + 2\lambda X^\top \left[ (AYY^\top AH) \odot \sigma'(XW_1) \right]$$

