# OpenReview forum: "The Geometry of Grokking: Norm Minimization on the Zero-Loss Manifold"
_ICLR.cc/2026/Conference — ICLR 2026 Conference Withdrawn Submission_

### Official Review · Reviewer_eqtw · 2025-10-18

**Soundness:** 3
**Presentation:** 2
**Contribution:** 2
**Rating:** 2
**Confidence:** 3

**Summary:**

The paper aims to establish a better theoretical understanding of the grokking phenomenon. They argue that gradient descent first reaches the zero-loss manifold (i.e., achieves perfect memorization) and then stays close to this manifold while decreasing the parameters’ norm, which eventually leads to generalization. Also, they provide a slightly modified learning dynamics where the post-memorization dynamics can be analyzed. They also provide some empirical results to support their claim.

**Strengths:**

Obtaining a better theoretical understanding of grokking is an interesting question that has attracted interest in recent years.  Explaining grokking by analyzing the dynamics after memorization is a natural approach, and, as the paper argues,  weight decay seems to play a key role in the post-memorization dynamics.

**Weaknesses:**

The bottom line is that I don’t think the contributions of the paper are significant enough.

In Section 4, the main contributions are Theorems 4.9 and 4.13. Theorem 4.9 provides a nice observation, namely, that with a small enough weight decay, once gradient flow reaches the zero-loss manifold, it will stay close to it. Then, Theorem 4.13 establishes that, under some assumptions, around the zero-loss manifold, the gradient of the unregularized loss will be roughly orthogonal to the manifold. Hence, the movement on the manifold seems to be mostly controlled by the weight decay. Theorem 4.13 considers the direction of the gradient of the unregularized loss, but when $\theta$ is close to the zero-loss manifold, the magnitude of the gradient is anyway very small, and hence, the weight decay becomes dominant. Also, the regularized loss might be small even if the distance to the manifold is large, and then the theorem does not apply. I view these results as nice observations, but not as a significant step towards a better understanding of grokking.

In Sections 5 and 6, the authors analyze a modified dynamics. In the case of two-layer networks, it corresponds to training the output layer much faster than the hidden layer. They obtain a closed-form expression of the output layer (as a function of the hidden layer) and of the dynamics of the hidden layer, and then show empirically (in Section 7) that delayed generalization also occurs under this regime. I think that it is an intriguing regime, but I don’t really understand how this discussion helps us understand grokking.

I did not verify the technical details.

**Questions:**

In Theorem 4.13, there is an assumption that  $proj_{\mathcal{Z}}$ $(S \cap \mathcal{Z}) \subseteq S$, but by definition if $x \in S \cap \mathcal{Z}$ then proj$_{\mathcal{Z}}(x) = x \in S \cap \mathcal{Z} \subseteq S$. I there a typo here?

---

> ### Author Response · Authors · 2025-11-14
>
> Thank you for your thoughtful review! We clarify several points that were misunderstood and highlight why our results provide substantive insight into the mechanics of grokking.
>
> > when $\theta$ is close to the zero-loss manifold, the magnitude of the gradient is anyway very small, and hence, the weight decay becomes dominant.
>
> There is an issue with this line of reasoning. The distance to Z depends on $\lambda$: smaller distance requires smaller $\lambda$. Hence, the strength of the gradient at $\theta$ will be of the same magnitude as the regularization. This is why it's essential to consider the angle of the gradient and why our theorem is necessary.
>
> > the regularized loss might be small even if the distance to the manifold is large, and then the theorem does not apply
>
> This is why we have the assumption of starting on (or near) Z. However, in overparametrized models, it's very unlikely (if not impossible) to have low loss without being very close to Z (think NTK regime).
>
> > I there a typo here?
>
> Yes, thank you.
>
> > I view these results as nice observations, but not as a significant step towards a better understanding of grokking.
>
> The novelty is precisely that we convert a set of folk intuitions into a formal, quantitative, and predictive framework for the entire post-memorization phase. We turn the informal post-memorization picture into a **mathematically rigorous, trajectory-level** description that yields **new, testable predictions** about grokking dynamics.
>
> > I think that it is an intriguing regime, but I don’t really understand how this discussion helps us understand grokking.
>
> The modified dynamics is not introduced as an unrelated curiosity. Its role is:
> (a) to isolate and simplify the dynamics of the representation layer that actually leads to grokking,
> (b) to produce a closed-form, analyzable trajectory for that layer using our constrained-optimization picture, and
> (c) to show that this simplified model reproduces delayed generalization and the structure of learned representations.
>
> Section 5 is exactly the tool that allows us to bring the geometric results from Section 4 into a concrete setting where the entire post-memorization dynamics can be written in closed form. Section 7 then shows that this closed-form prediction matches the behavior of real grokking networks, demonstrating that the “norm minimization on Z” mechanism is not only intuitive but predictive.
>
> Thus, Sections 5-7 are the bridge from the abstract theorems to actual grokking behavior.
>
> In light of these clarifications, we kindly invite the reviewer to reconsider their evaluation, including the confidence level, if they feel that the clarified positioning and contributions change the strength or significance of the paper.

---

> > ### Comment · Reviewer_eqtw · 2025-11-21
> >
> > > This is why we have the assumption of starting on (or near) Z. However, in overparametrized models, it's very unlikely (if not impossible) to have low loss without being very close to Z (think NTK regime).
> >
> > Even if you start near Z, the distance from Z might increase during training without increasing the regularized loss (think about a very flat optimization landscape).
> >
> >
> > Overall, I thank the authors for their response, but I maintain my evaluation.

---

> ### Author Response · Authors · 2025-11-21
>
> > Even if you start near Z, the distance from Z might increase during training without increasing the regularized loss (think about a very flat optimization landscape).
>
> This is true and your example is great. But we want to emphasise that, with a small enough $\lambda$, one can guarantee that an arbitrarily small distance to Z is reached. Using the NTK regime, one can show that for any initialization and desired distance $\epsilon$, there is a corresponding small $\lambda > 0$ such that the model is guaranteed to reach $\mathrm{dist}_{\mathcal{Z}}(\theta(t)) < \epsilon$. To get an intuition for why this is true, think of how an over-parametrized model with $\lambda = 0$ will always converge to Z.
>
> Combining the above fact with the proof for our Theorem 4.8 (Stability of Z) yields the fact that a model with arbitrary initialization becomes provably constrained arbitrarily close to Z during the "grokking" phase for a small enough $\lambda > 0$, thus removing the need for our assumption of starting exactly on Z.
>
> We did not include this proof in our paper because we wanted to understand what happens after memorization, not until memorization. Moreover, the pre-memorization regime (aka NTK regime) is quite well studied in the existing literature. Hence, we believe that starting exactly on Z is a reasonable assumption. However, we are happy to extend our work if the reviewer finds it necessary.
>
> Would the reviewer change their evaluation if we added a full proof in the paper for the above statement **OR** if we found an existing proof in the literature for exactly this statement?
>
> Thank you!

---

> > ### Comment · Reviewer_eqtw · 2025-11-23
> >
> > Grokking means that the model first reaches Z (or close to it) and then, after many iterations, it also starts showing non-trivial generalization. The theoretical result does not show that GD reaches Z long before generalization begins (it does not rule out the possibility that it reaches Z simultaneously with the improvement in generalization). The general fact that the dynamics of GD continues due to the weight decay even after reaching Z is obvious, and I don't see how the specific quantitative result in the paper helps in understanding grokking. So, unfortunately, I tend to keep my current score.

---

### Official Review · Reviewer_CkgN · 2025-10-29

**Soundness:** 2
**Presentation:** 1
**Contribution:** 2
**Rating:** 2
**Confidence:** 4

**Summary:**

The manuscript claims that post-memorization dynamics is driven by weight decay. Explicitly, that after memorization the learning trajectory is confined to the zero-loss manifold and weight decay minimizes the weight norm subject to the zero loss constraint. The authors claim that this is cause of grokking.

**Strengths:**

The general idea -- that in the presence of weight decay dynamics on the zero loss manifold are determined by the weight decay term -- is (trivially) correct.

**Weaknesses:**

The actual theorems seem trivial to me, and the presentation is over complicated. If one assumes that the dynamics has reached the zero loss manifold, then by definition further dynamics are not controlled by loss gradients (which vanish) but only by the regularization, if present. However, this does not answer two things which are crucial for the proposed mechanism to work:
1. why should the dynamics reach the zero loss manifold at all, in the presence of regularization?
2. Assuming it does,  why/when/under which conditions/etc does should weight norm minimization on this manifold should lead to better generalization?

Sec 5.2 is poorly written and hand wavy, and contains unsupported claims and trivial mistakes. For example, the claim that "it is highly unlikely that our trajectory will pass through the same θ1 twice" is loose and ill defined, and does not imply Eq. 9

In addition, the manuscript is over complicated with its non-standard definitions ("available direction"=tangent vector etc).

**Questions:**

none

---

> ### Author Response · Authors · 2025-11-14
>
> Thank you for you review! We address your concerns below.
>
> > If one assumes that the dynamics has reached the zero loss manifold, then by definition further dynamics are not controlled by loss gradients (which vanish) but only by the regularization
>
> Note that even if the model is exactly on the zero loss manifold (Z) at one point in time, that does not guarantee that it will stay there. This is because the regularization is free to push it out of Z (unless the regularization is perfectly tangent to Z, which is highly unlikely). Kindly, please take a few minutes to think about why this is the case. It appears that your criticism is based on an incorrect understanding.
>
> It is not trivial to show that the model remains near Z after reaching it once. This is what we achieve by proving Theorem 4.8 (Stability of Z). Then it is not trivial to show that the gradient near Z does not induce any movement in the tangent space. This is what we show by Theorem 4.14 (Gradient Orthogonality). We have added a new toy model visualization in Section 3.4 to give an intuition for the statement of Theorem 4.14. We kindly ask you to read it.
>
> > why should the dynamics reach the zero loss manifold at all, in the presence of regularization?
>
> Indeed, it is possible that the model never arrives exactly on Z (i.e. it never achieves exactly zero loss). However, the gradient orthogonality only requires the model to be withing a small distance $\epsilon$. The assumption of an initialization on Z is not necessary, but it greatly simplifies the proofs. Otherwise, we would need an additional theorem that an arbitrarily small distance to the loss is achieved by some small regularization term. Proving this would require further assumptions on the model (e.g. Neural Tangent Kernel, etc).
>
> > the claim that "it is highly unlikely that our trajectory will pass through the same θ1 twice" is loose and ill defined, and does not imply Eq. 9
>
> We motivate this assumption based on the comprehensive literature on multiple points of stochastic processes. For example, even with a dimensionality as small as d ≥ 4, it is known that a Brownian motion in Rd contains no self-intersections almost surely (Morters & Peres (2010), Chapter 9). More generally, Dalang et al. (2021) prove the non-existance of multiple points for a wide class of Gaussian random fields. Of course, one can come up with specific pathological loss functions where collisions are inevitable, but we expect our assumption to hold for neural networks with standard architectures and datasets given their high dimensionality and stochasticity.
>
> > the manuscript is over complicated with its non-standard definitions ("available direction"=tangent vector etc).
>
> We have replaced "available direction" with "tangent direction". Thank you for this suggestion! We are happy to hear any other possible improvements.
>
> In light of these corrections, we kindly ask you to reconsider your score and/or confidence.

---

> > ### Comment · Reviewer_CkgN · 2025-11-16
> >
> > > Note that even if the model is exactly on the zero loss manifold (Z) at one point in time, that does not guarantee that it will stay there....
> >
> > This is indeed not trivial, but the authors assume vanishing weight decay (Assumption 4.6), in which case the gradients due to the loss will, by definition, dominate the regularization gradient.
> >
> > > We motivate this assumption based on the comprehensive literature on multiple points of stochastic processes...
> >
> > It is of course true that Brownian motion in large dimensions is non-recurrent, but as far as I understand this has nothing to do with the current discussion, which is about deterministic gradient-flow dynamics.
> >
> > But this is really just nitpicking. I am willing to accept the very reasonable assumption that the trajectory does not step on the same $\theta_1$ twice. Much more importantly - even in this case, **this does not imply Eq. (9).** In fact, $\phi$ is only defined for a 1-dimensional curve in $\theta_1$ space, and is generally not a function from $\mathbb{R}^{d_1}$ to $\mathbb{R}^{d_2}$, like the authors wrongly claim in the text. If such a function is indeed defined also for points not visited by the 1D trajectory during one specific training, I did not see such a definition in the manuscript or in the appendices.
> >
> > Therefore, since $\phi$ is not even defined outside this curve, it cannot be differentiated in directions orthogonal to this curve, and in turn Eq. (9) is ill defined since it requires the gradient of $\phi$ in an arbitrary direction.

---

> ### Author Response · Authors · 2025-11-16
>
> Thank you for your thoughtful response! We respond to your concerns below.
>
> > This is indeed not trivial, but the authors assume vanishing weight decay (Assumption 4.6), in which case the gradients due to the loss will, by definition, dominate the regularization gradient.
>
> There is an issue with this line of reasoning. The distance to Z depends on $\lambda$: smaller distance requires smaller $\lambda$. Both the gradient and the regularization vanish as $\lambda$ vanishes. Hence, it is not true that one will dominate the other in the limit since the strength of the gradient at $\theta$ will be of the same magnitude as the regularization. This is why it's essential to consider the angle of the gradient and why our theorem is necessary. Please consider reading the newly added Section 3.4 to get an intuition of our theorem.
>
> > I am willing to accept the very reasonable assumption that the trajectory does not step on the same $\theta_1$ twice. Much more importantly - even in this case, this does not imply Eq. (9).
>
> Thank you for clarifying your concerns! Indeed, for eq (9) we do not say what the function looks like outside of the training trajectory. We just assume the existence of a function $\phi : R^{d_1} \rightarrow R^{d_2}$ with the property that $\theta_2 = \phi(\theta_1)$ along the training trajectory. Note, however, that is not an issue for Eq. 9, which is just the gradient of the loss (eq. 7) in a point on the training trajectory, with $\phi(\theta_1)$ swapped in for $\theta_2$. Hence, eq. 9 does not require any additional properties of $\phi$ besides eq. 8.
>
> We only require differentiability of $\phi$ later on, in Sec. 5.1, but by that time we have specified an approximation of $\phi$ that is defined on the entire domain. We also state our assumption that $\phi$ is differentiable in Thm 5.1.
>
> We will add a statement in the camera-ready version that Eq. 9 holds only on the training trajectory where $\phi$ is always defined. We hope these explanations clarify how our derivations are well defined and fully rigorous. We would be very happy to hear if the reviewer has any suggestions to improve the clarity of our work and reduce the risk of similar misunderstandings in the camera-ready version.
>
> > Assuming it does, why/when/under which conditions/etc does should weight norm minimization on this manifold should lead to better generalization?
>
> Regarding this concern from the original response, we want to emphasise that answering this question is only possible for a specific architecture and dataset. A great example is the work of [1] who show that max-margin solutions of a two-layer net with quadratic activations trained on group operations learn fourier features. However, in our work, we study the training dynamics in the post-memorization regime generally, hence it is not possible to give specific generalization guarantees. It is fully possible that the post-memorization dynamics do not induce generalization, as is in fact shown by [2] (a.k.a. misgrokking).
>
> In light of these clarifications, we kindly ask the reviewer to reconsider their score and/or confidence. Otherwise, we are happy to clarify any remaining concerns.
>
> [1] Morwani, Depen, et al. "Feature emergence via margin maximization: case studies in algebraic tasks." arXiv preprint arXiv:2311.07568 (2023).
>
> [2] Lyu et al., "Dichotomy of Early and Late Phase Implicit Biases Can Provably Induce Grokking." ICLR 2023

---

### Official Review · Reviewer_4cYX · 2025-10-29

**Soundness:** 2
**Presentation:** 4
**Contribution:** 3
**Rating:** 4
**Confidence:** 4

**Summary:**

The authors prove that once a grokking model reaches the zero-loss set, its subsequent trajectory can be described as minimizing the weight norm subject to the constraint of remaining on the zero-loss set (under a set of suitable assumptions). They show empirically that a toy model that simulates these constrained dynamics behaves qualitatively similarly to grokking models (exhibiting the characteristic phenomena of delayed generalization and circular representation learning).

**Strengths:**

**Crisp, elegant result:** The overall takeaway is quite elegant and easy to understand.

**Very well-written**: This was an exceptionally clear paper. I was really impressed with how easy this was to read. Similarly, the figures were wonderfully designed.

**Interesting methods**: The separation of dynamical systems into slow variables and fast variables that can be integrated out is a pillar of dynamical systems analysis. Applying it in this setting to understand learning on the zero-loss set is quite creative and interesting. I think this is likely to be a productive set of tools to apply to study other toy models and interesting dynamical phenomena. I want to see this paper published for this reason.

**Weaknesses:**

**1. Missing empirical validation.** After solving for the closed-form expression of the post-memorization dynamics of (part of) a grokking model, the authors show that simulating this training process reproduces two of the behaviors associated with grokking: delayed generalization and circular representation learning. But, if I understand correctly, this does not establish the central theoretical claim (lines 113-118):

> While the previous example illustrates our theoretical framework, it does not capture its full generality. It is unsurprising that applying weight decay encourages a reduction in norm. However, the central claim of this paper is significantly stronger: we argue that the learning dynamics under weight decay do not merely follow some norm-decreasing direction, but rather evolve along the direction *that optimally minimizes the norm, subject to remaining on the zero-loss manifold*.

The authors have not established that delayed generalization and representation learning occur *only* in the case where the training trajectory optimally minimizes the norm subject to remaining on the zero-loss manifold. These two observed phenomena may still be compatible with many other (suboptimal) learning trajectories.

On a closer reading, I believe that most of the current writing does not make any false claims about what these experimental results imply. For example, the following lines do not need to be changed:
- Lines 022–023: "confirm that simulating the training process ... reproduces both the delayed generalization and representation learning..."
- Lines 427–428: "validate that our approximate learning dynamics reproduce the phenomena observed during standard training"

However, there are a few instances that can be interpreted too broadly and thus should be rewritten:
- Line 062: "we validate our theoretical insights"
- Line 375: "we empirically validate our combined theoretical insights"
- Lines 468–470: "formally established that the learning dynamics ... approximate as the minimization of the weight norm within the zero-loss set."

Minimum correction: In addition to addressing the above overstatements, the main body should explicitly comment on the remaining discrepancy between theoretical predictions and empirical confirmation. Omitting this would implicitly suggest stronger empirical confirmation than is currently demonstrated.

Full correction: Convincingly establishing that the true training trajectory actually does follow the zero-loss-constrained norm-minimization path requires additional experiments. For example, you could test the alignment (via cosine similarity) between actual updates and predicted updates under training. Alternatively, you could argue for the necessity of the optimal direction if you can show that other learning trajectories within the zero-loss set do not produce these behaviors (or perhaps much more slowly). These are just suggestions. I'm sure the authors can come up with better ways to validate this.

**2. False assumption about negligible singularities.** The manuscript claims that singular points "form a null set" and that the "probability of encountering a singularity during standard training is exactly zero" (lines 157–158), and then assumes the trajectory never passes through them (Assumption 4.7). There is an additional assumption about a constructed set not containing singularities in Theorem 4.13. I believe that the current conditions are insufficient for these statements to hold and that they are partially false as written.

2.1 Assumption 4.7:
> We assume that our training trajectory does not pass through singular points. This is motivated by the fact that singular points form a null set, i.e. a set of Lebesgue measure zero.

The first problem is that, as written, smoothness alone does not imply this set is measure zero. That requires invoking an additional non-degeneracy condition (that there is at least some parameter with full rank) to argue that the rank-deficient locus is "thin." I think these concerns would be resolved by including some wording about "standard genericity conditions."

However, I believe there may still be a problem in that standard architectural degeneracies (e.g., $ReLU(\alpha x) = \alpha ReLU(x)$ for $\alpha> 0$) would invalidate these conditions. Since these degeneracies also hold off of the measure-zero set, I do not believe they pose a substantial challenge to the theoretical argument and can be easily eliminated.

The second problem is that the stated assumption does not follow from the provided motivation. The fact that singular points form a null set is sufficient to establish that *before* training (under a random initialization), the probability of encountering a singularity is vanishing. However, the probability *after* training can, in general, be non-zero that a 1D training path crosses such measure-zero sets at isolated times. This claim can be strengthened by arguing theoretically that the path remains in a compact subset bounded away from the singularities and/or with empirical evidence showing that, e.g., the smallest singular value of $\nabla \mathcal F$ is non-zero.

2.2 Theorem 4.13:
I believe you need a stronger statement than the absence of singularities (line 673) and that you need an additional statement that says something about keeping a uniform gap between singularities and the training trajectory. From my understanding, if the training trajectory is allowed to get arbitrarily close to singularities, the positive constant $\lambda_\text{inf}$ introduced in lines  721–723, can get arbitrarily small, which renders the resulting bound in Theorem 4.13 vacuous. (That the training trajectory can get arbitrarily close to singularities is no surprise. After all, the entire paper rests on the condition that the training trajectory gets close to the measure-zero zero-loss set.)

Minimum correction:
My first concern (2.1) would be resolved by including some wording about "standard genericity conditions", clarification from the authors about my question on standard degeneracies, and weakening the statement about the loss trajectory avoiding singularities or providing additional theoretical/empirical evidence that the trajectory does not pass near singularities.

My second concern (2.2) would be resolved by including an additional assumption on the gap between singularities and the training trajectory, or perhaps simply clarification from the authors on this point. I am less confident with (2.2) than with (2.1).

Full correction:
I am quite confident that the authors' general attitude towards singularities being negligible is false. Neural networks have incredibly rich degeneracies, and a lot of well-established theory (notably Watanabe's singular learning theory) emphasizes the key role these singularities play in learning dynamics. If the core theoretical derivation of the paper could be strengthened to drop the assumptions of negligible singularities (which seems very difficult to me), then this would resolve my remaining concerns.

I recommend acceptance conditional on (i) softening the empirical‑validation language and acknowledging the remaining gap, and (ii) qualifying the singularity statements (ideally with a uniform‑gap assumption or an empirical singular‑value check). Addressing both minimum fixes moves me to accept; adding one of the maximum fixes would push me towards a strong accept.

**Questions:**

Do you see any connections to existing work within the literature on inductive biases on margin-maximization?

See the questions I've interspersed among my concerns under weaknesses.

---

> ### Author Response · Authors · 2025-11-14
> **Full correction of weaknesses**
>
> We would like to thank you for the very comprehensive, highly insightful, and very encouraging review!
>
> We believe that your concerns are completely valid. We want to thank you for pointing out the two weaknesses and proposing solutions.
>
> We believe that the new manuscript addresses **both** of the highlighted weaknesses in **full**.
>
> **Missing empirical validation:** we added new gradient similarity experiments on a modular addition task, showing that gradient similarity is very high during the grokking phase. Please see the new Section 4.6.
>
> **False assumption about negligible singularities:** We added a new theoretical result, Theorem 4.10 (Regularity of Z), showing that Z is completely regular for almost all datasets. Furthermore, we have weakened the assumption in the core theorem by requiring no singularities only in $S \cap Z$, not all of $S$. This ensures that gradient orthogonality holds for almost all datasets. No changes were needed for the proof, as it was already dependent only on singularities in $S \cap Z$.
>
> If our fixes fulfill your criteria for a double full correction, we would like to kindly ask you to reconsider your score and confidence. If not, we would be very happy to hear any outstanding criticisms.

---

### Official Review · Reviewer_1pPH · 2025-10-31

**Soundness:** 3
**Presentation:** 3
**Contribution:** 2
**Rating:** 2
**Confidence:** 3

**Summary:**

This paper analyzes the grokking phenomenon and argues that it can be explained through the lens of constrained optimization: fast memorization corresponds to fast convergence to the zero-loss manifold, then weight decays drives the model to minimize the weights' norm and thus (slowly) travel on the zero-loss manifold, towards a generalizing solution. Going further, the paper shows that this dynamic can be analyzed by decoupling the embedding layer from the rest of the network.

The theoretical results are crystallized in Thm 4.9, showing that optimization remains close to the zero-loss set Z, once reached, Thm 4.13, showing that gradient of the loss is orthogonal to Z in the vicinity of Z, suggesting that the trajectory is only driven by weight decay when close to Z / in Z, and Thm 5.1 which is essentially a technical tool allowing the authors to analyze the training dynamics of a subset of the parameters in isolation. This last theorem is used to provide an analysis of the first layer dynamic in a two layer network.

Experiments are provided to validate the insights of Thm 4.9 and 4.13 and the simulated dynamics of the 2 layer network.

**Strengths:**

- The paper is overall well written and easy to follow.
- The topic is definitely timely and relevant and the analysis is interesting. In particular, the approach taken to study isolated dynamics could be relevant in other settings.

**Weaknesses:**

- Some very relevant papers are missing for the literature review. In particular, Q1 investigated in this paper has already been quite thoroughly addressed in [1]. [2] also shows that this behaviour extends to other form of regularization, beyond weight decay. While the technical tools used in this submission to support the hypothesis that grokking is an artifact of regularization (memorization induced by data fitting term followed by slow convergence to generalizing solution driven by regularization) seems at first sight different from the one used in [1] and [2], this potentially affects significantly the novelty of this submission (at least the parts related to Q1, i.e. Section 3 & 4, Thms 4.9&4.13). It is important that the authors clarify during the rebuttal the position of the current submission compared to [1] and [2], and to check references in [1] and [2] as there may be additional relevant papers that were missed in the literature review.  [My low score could be updated during rebuttal depending on how this point is addressed by the authors].

- Overall, the results presented in Thm 4.9 and 4.13 are interesting but somehow not too surprising:
   - Thm 4.9 boils down to observing that the regularization term can be taken to be arbitrary small to make the minimum arbitrary close to any point in Z, and thus constraining the dynamics to stay arbitrary close to the zero-loss set.
   - Thm 4.13 states that gradients points towards the zero-loss set in its vicinity, which as far as I see is quite expected. I am not entirely convinced that this theoretical result is strong enough to derive the conclusion the authors get to in Rmk 4.14.

- Section 6 is interesting but seems to be more hastily written than the previous ones. It is difficult to get to the high-level message that this section wants to convey. It shows an interesting derivation, but lacks commenting on its relevance in the current scope from a high-level point.

[1] Lyu et al., "Dichotomy of Early and Late Phase Implicit Biases Can Provably Induce Grokking." ICLR 2023

[2] Notsawo et al., "Grokking Beyond the Euclidean Norm of Model Parameters." ICML 2024

**Questions:**

- Please situate your work w.r.t. to refs [1] and [2]. In particular, how does the analysis presented in [1] already supports the conclusions of the analysis presented in Section 3 and 4?

- Can you clarify how Thm 4.13 goes further than simply observing that gradient is zero in Z, hence dynamics are only driven by weight decay?

#### Minor points / questions

- Thm 4.13: why do we need the assumption $\mathrm{proj}_Z(S\cap Z) \subset S$? Can you give an example where this would not be satisfied?

- Thm 4.13: Why are 2 constants, C and x_0,needed? Couldn't you "absorb" the constant C in x_0? I.e. by defining $\tau = Cx_0$, we can get $\tau dist_Z(\theta)$ as the rhs of ineq (6), no?

- Eq 19 and Eq 20: both $(H^TH)^{-1}$ and $(HH^T)^{-1}$ appear in these equations but only one of the two is invertible! (unless H is square...).

- Eq 20: Why doesn't terms related to the loss L appear in this equation? It seems to me that $L(W_1,\phi(W_1))$ is not necessarily $0$ for all choices of $W_1$...

---

> ### Author Response · Authors · 2025-11-14
>
> We would like to thank you for your insightful review and suggestions!
>
> > Please situate your work w.r.t. to refs [1] and [2]. In particular, how does the analysis presented in [1] already supports the conclusions of the analysis presented in Section 3 and 4?
>
> Thank you for referring us to [1] and [2]. We believe that our results are orthogonal to theirs.
>
> [1] is related but it seems that their setup is mutually exclusive with ours. Their paper studies grokking in the limit of a very large initialization $\| \theta \| \rightarrow \inf$, whereas we use the limit of a vanishing weight decay $\lambda \rightarrow 0$. Their paper has two theoretical results about the two regimes: lazy and rich. The learning dynamics in the lazy regime (memorization) are explained by the NTK, but this falls outside the scope of our paper, hence there is no overlap in this regard. Their results about the rich regime are more resemblant of our work, but there are fundamental differences. They establish convergence to a max-margin solution in the limit of infinite norm initialization. In this regard, our results are arguably stronger, since we can explain grokking from a bounded initialization (albeit we require vanishing weight decay, which could be seen as a trade-off). Moreover, to the best of our understanding, the proofs in [1] use weight decay as merely a weight reduction mechanism. Our work shows that weight decay can play a much more complex role, inducing highly non-linear training dynamics.
>
> [2] shows that grokking can emerge from other types of regularization besides the euclidean norm. This is again orthogonal to our paper, since our theoretical results could be trivially generalized to other types of regularizations if we maintain the assumption of a vanishing regularization coefficient. Any regularizer that is sufficiently smooth can be plugged into the same framework to obtain the corresponding constrained-optimization dynamics on Z (we will make this point explicit in the camera-ready).
>
> To the best of our knowledge, norm minimization on the zero-loss manifold is a novel idea. We are not aware of any works that study (or even mention) this behaviour.
>
> > Can you clarify how Thm 4.13 goes further than simply observing that gradient is zero in Z, hence dynamics are only driven by weight decay?
>
> This seems to be a common point of confusion. In order to clarify the theorem statement, we have added another visualization with a toy model in Section 3.4. Note that the fact that the gradient is zero in Z is not sufficient, as the model will not find itself exactly in Z (except at initialization). Training dynamics take place at a distance $\epsilon$ from Z, as we establish in Thm 4.8. We prove that gradient orthogonality holds uniformly over this entire region, not just in Z. Please read the new Section 3.4.
>
> > Thm 4.13: why do we need the assumption $\mathrm{proj}_Z(Z \cap S) \subset S$? Can you give an example where this would not be satisfied?
>
> Indeed, this was a typo. We meant to say $\mathrm{proj}_Z(S) \subset S$. Thank you for pointing this out!
>
> > Thm 4.13: Why are 2 constants, C and x_0,needed?
>
> That is a great suggestion! Thank you very much! We have integrated it by adding an additional small step to our proof.
>
> > Eq 19 and Eq 20 [...]
>
> Thank you for pointing out this lack of clarity! We have fixed it.
>
> > Eq 20: Why doesn't terms related to the loss L appear in this equation?
>
> That section is based on the assumption that the second layer can achieve zero-loss on its own. This is almost always true if $d_h > n$ (i.e. a strongly overparametrized model with more hidden units than training samples). This is also what allows us to write $H^+ = H^\top (H H^\top)^{-1}$, since $H \in \mathbb{R}^{n \times d_h}$. We have clarified this in the revised version. Thank you for pointing this out.
>
> > Eq 20: Why doesn't terms related to the loss L appear in this equation?
>
> If our answers and additions are satisfactory, we kindly ask you to reconsider your score.

---

> > ### Comment · Reviewer_1pPH · 2025-11-21
> >
> > Thank you for your answers, fixing the typos and integrating my comments in the revision. I believe the positioning of your work with respect to Ref. [1] and [2] mentioned above should be discussed in the revision.

---

### Author Response · Authors · 2025-11-14

Thank you again for your constructive feedback. We have revised the paper and made several substantial improvements based on your comments:
- **New empirical results (Section 4.6):** We added gradient-similarity experiments on modular addition, showing that during the grokking phase the update direction aligns extremely closely with the predicted norm-minimizing direction on the zero-loss manifold.
- **New theoretical result (Theorem 4.10):** We prove that the zero-loss set Z is completely regular for almost all datasets. This allows us to weaken the core assumption in Theorem 4.14: we now require the absence of singularities only in $S \cap Z$, which holds almost always. The original proof already depended only on this restricted condition, so no changes were needed.
- **New visualization (Section 3.4):** We added an improved toy-model figure that gives immediate geometric intuition for gradient orthogonality near Z.
- **Clarity and correctness improvements:** We fixed typos, clarified technical statements, and streamlined several proofs and explanations.


We hope these changes address your concerns and make the contributions of the paper clearer. We would be grateful if you could take them into account when updating your assessment.

---

### Comment · Area_Chair_iwSU · 2025-11-17
**Please read and reply to authors' responses**

Hi,

I know that some of you are probably busy with rebuttals for your own ICLR submissions, but please be sure to read the authors' responses to your initial reviews and take part in the discussion.

Best,\
AC

---

### Note · Authors · 2026-01-04

I have read and agree with the venue's withdrawal policy on behalf of myself and my co-authors.